# Harnessing protein folding neural networks for peptide–protein docking

Tomer Tsaban [1,2], Julia K. Varga[1,2], Orly Avraham[1,2], Ziv Ben-Aharon [1], Alisa Khramushin[1] & Ora Schueler-Furman [1✉]

Highly accurate protein structure predictions by deep neural networks such as AlphaFold2 and RoseTTAFold have tremendous impact on structural biology and beyond. Here, we show that, although these deep learning approaches have originally been developed for the in silico folding of protein monomers, AlphaFold2 also enables quick and accurate modeling of peptide–protein interactions. Our simple implementation of AlphaFold2 generates peptide–protein complex models without requiring multiple sequence alignment information for the peptide partner, and can handle binding-induced conformational changes of the receptor. We explore what AlphaFold2 has memorized and learned, and describe specific examples that highlight differences compared to state-of-the-art peptide docking protocol PIPER-FlexPepDock. These results show that AlphaFold2 holds great promise for providing structural insight into a wide range of peptide–protein complexes, serving as a starting point for the detailed characterization and manipulation of these interactions.

[1] Department of Microbiology and Molecular Genetics, Institute for Biomedical Research Israel-Canada, Faculty of Medicine, The Hebrew University of Jerusalem, Jerusalem, Israel. [2] These authors contributed equally: Tomer Tsaban, Julia K. Varga, Orly Avraham. ✉email: ora.furman-schueler@mail.huji.ac.il

Peptide–protein interactions are highly abundant in living cells and are important for many biological processes[1]. It is estimated that up to 40% of interactions in cells are mediated by peptide–protein interactions, or peptide-like interaction:[2] short segments, isolated or embedded within unstructured regions that mediate binding to a partner[3]. In addition, peptides are often used for biotechnological applications, drug delivery, imaging, as therapeutic agents, and other applications[4,5], by binding proteins and mediating or blocking interactions.

Determining the 3-dimensional structure of these peptide–protein complexes is an important step for their further study. They can provide the basis to identify hotspot residues that are crucial for binding[6–8], and by mutating these hotspots, the functional importance of a given interaction can be uncovered[9]. They could help to better understand disease-causing mutations and also serve as a starting point for the design of strong and stable peptidomimetics[10,11].

However, peptide-mediated interactions pose significant challenges, both for their experimental as well as their computational characterization: These interactions are in many cases weak, transient, and considerably influenced by their context, resulting in often noisy experiments. Widely used structure determination methods (e.g., X-ray crystallography) are not applicable to many of these interactions. Computational modeling, and particularly blind peptide–protein docking[12], is hindered by the lack of known structure for the peptide side, in contrast to classical domain-domain docking, where the structure of the free individual domains is usually defined. In order to succeed in the study and design of peptide–protein interactions, we must gain a better understanding of the peptide conformational preferences.

One way to approach this challenge is based on the observation that a peptide bound conformation is often present in solved monomer structures[13]. Based on this finding, we developed the high-resolution blind peptide docking protocol, PIPER-FlexPepDock (PFPD)[13]. First, a representative ensemble of fragments is extracted from monomer structures using the Rosetta Fragment Picker[14], which takes into account both sequence and (predicted) secondary structure similarity. Then this ensemble is rigid-body docked onto the receptor with the PIPER protocol[15], followed by short local refinement by Rosetta FlexPepDock[16], which simultaneously optimizes internal peptide and rigid-body degrees of freedom. Numerous other peptide docking approaches have since been developed[12,17], many focusing on efficient low-resolution docking[18,19], others leveraging information about protein interfaces to find matches for similar interface patches[20–22].

Another way to approach the global peptide docking challenge is to view the binding of a peptide to its partner as the final step of protein folding, complementing the receptor surface with a missing piece[23]. Indeed, functional proteins can be reconstituted experimentally from short fragments of the original sequence, indicating that covalent linkage is not necessarily a prerequisite for monomer folding[24,25]. We and others have successfully modeled peptide–protein interactions using this principle, by finding fragments in monomer structures and on protein-protein interfaces that could complement structural patches derived from the surface of a given receptor[20–22,26]. These concepts lay the groundwork for novel approaches in peptide–protein docking, where the vast information inherently stored in folded monomer structures is efficiently integrated in the search space for peptide docking.

The advances in the field of protein structure prediction in recent years open up exciting opportunities to fully leverage such information. The development and application of deep learning (DL) neural network (NN) architectures to predict monomeric protein structures provided us with highly accurate computational models as particularly showcased by the last CASP14 experiment[27]. AlphaFold2 (AF2) developed by Google Deepmind was able to generate models of exceptional accuracy, approaching the resolution of crystallography experiments[28]. Significantly improved modeling was also reported for RoseTTAFold, developed by RosettaCommons, that followed ideas from AF2 and also implemented fully continuous crosstalk between 1D, 2D and 3D information[29]. Most importantly, AF2, as well as RoseTTAFold, are now freely available to the scientific community[30,31], opening up powerful avenues for protocol development and applications to many biological systems that were not amenable to structural characterization in the past. These are truly exciting times!

Can such NNs also model peptide–protein interactions, and not only monomers? If peptide–protein interfaces are indeed abundant in monomer structures, and if indeed peptide–protein interactions can be captured as protein folding as stated above, RoseTTAFold and AF2 should, in principle, also allow for the modeling of peptide–protein complex structures. Moreover, they could alleviate the lack of data impairing the ability to fully employ DL for peptide–protein interactions. We note that both RoseTTAFold and AF2 NNs were trained on single chain protein structural data, and both use Multiple Sequence Alignments (MSA) as a critical step in structure prediction. Prediction of protein-protein complexes was shown to be possible given an informative MSA[27,29,32], and it has also been explored whether it is indeed necessary to provide paired sequences for successful extraction of interface information[33,34]. As both methods heavily rely on good quality MSA, the main challenge would be to accurately predict the peptide conformation. Mainly due to their short length, creating an effective MSA for these regions is challenging.

Here we present a global peptide–protein docking approach that incorporates the biological concept of peptide–protein interactions mimicking protein folding and harnesses NNs trained to predict monomeric protein structures. We show that by connecting the peptide to the receptor (e.g., by a poly-glycine linker), monomer folding NNs generate accurate peptide–protein complex structures (a similar idea was proposed in parallel by others[35]). This is possible thanks to the ability of AF2 to (1) accurately identify unstructured regions[36] and model these as extended linkers, and (2) predict peptide-receptor complexes without a multiple sequence alignment for the peptide partner, as we demonstrate in this study. Best performance is obtained by combining our linker-based strategy with modeling of peptide–protein complexes by presenting two separate chains to AF2. The latter has been implemented for the modeling of homo- and hetero-multimers in several recent studies on AF2[36,37].

We perform a short calibration on a small representative, previously well-studied set of protein-peptide interactions, consisting of peptides with and without known binding motifs[13]. We then provide a detailed comparison to the currently top-performing global peptide docking protocol PFPD[13]. We then assess the protocol on an extensive, non-redundant set of curated peptide–protein complexes consisting of 96 interactions, each involving a distinct fold. Finally, we explore specific types of interactions of special interest, including examples in which peptide binding induces a large conformational change in the receptor upon binding. The latter are very challenging to model using docking, but easily amenable to AF2 which models the complex as a whole. Beyond presenting an approach to dock peptides, this study provides another view on what AF2 may have learned beyond memorization.

## Results

**Adapting NN-based structure prediction to peptide docking.** By adding the peptide sequence via a poly-glycine linker to the C-terminus of the receptor monomer sequence, we mimicked

peptide docking as monomer folding. This is based on the assumption that the NN should identify the poly-glycine segment as non-relevant and use it merely as a connector (Fig. 1a). In contrast to AF2, a similar tactic using RoseTTAFold did not succeed but rather attempted to fold the polyglycine into a globular structure or create various loops with intra-loop interactions (Supplementary Figure 1). This can be explained by the fact that RoseTTAFold was not trained to identify unstructured regions[29], in contrast to AF2 where these regions were not removed before training. We, therefore, proceeded only with AF2 for NN-based peptide docking.

**AF2 predicts peptide–protein structures at high accuracy**. We evaluated the feasibility of our approach on a set of peptide–protein complexes described in a previous study (26 complexes, 12 of which have an experimentally characterized peptide binding motif, termed motif and non-motif sets)[13]. Figure 1a shows an example of accurate modeling, and another example where AF2 fails. The failure is easily identified by the poly-glycine linker "throwing" the peptide segment into space. Overall, AF2 models 75% of the interactions in the motif set within an impressive 1.5 Å RMSD, while performance is inferior for the non-motif set (36% within 2.5 Å RMSD) (Fig. 1b, upper left panel; RMSD calculated over the peptide interface residue backbone/heavy atoms. upper right panel: corresponding RMS values calculated over the whole interface after its alignment, corresponding to the CAPRI Irms measure, see Supplementary Fig. 2 and Methods for details).

We were able to obtain these results after only minor optimization of the default AF2 monomer structure prediction protocol for peptide docking (see Methods). Most importantly, we also modeled the interaction with separate chains, as has already been suggested for protein docking[33,34,37]. This implementation provided complementary results (see Supplementary Fig. 3; results are detailed in Supplementary Data 1). We, therefore, merged both approaches and assessed performance based on the best RMSD model among 10 generated models (i.e., five linked and five separate chain models). Besides the type of linkage, we evaluated several other parameters that could affect performance (see Supplementary Fig. 4). Increasing the number of recycles from 3 to 9 resulted in slightly better performance. Therefore we continued using 9 recycles for subsequent runs.

Our results demonstrate that even without any dedicated training, AF2 predicts accurate models at a good resolution for a high fraction of interfaces. This prediction is possible despite the lack of informative MSAs for the peptide partner, and therefore of corresponding co-evolutionary signals between the peptide and the receptor. This lack is expected as we provide an input that is fragmented, *i.e.* an artificial fusion or a segment too short to yield significant alignments.

Performance for the motif set is notably better compared to PFPD (where we select the best RMSD model among the top 10 cluster centers, as reported previously[13]), while PFPD performs slightly better for interactions with no reported motif (the non-motif set) (Fig. 1b). Importantly, PFPD and AF2 results fail on different examples (Fig. 1c), indicating that a future combination of the two approaches may boost performance even further.

In contrast to PFPD, using AF2 for peptide docking includes modeling of both the peptide and the receptor. The performance calculated over the full interface (i.e., interface residues of both peptide and receptor, Fig. 1b, right) is similar to the one of the peptide, thanks to highly accurate modeling of the individual receptor as well as the peptide structures (Fig. 1d). A non-trivial insight is that accurate modeling of the individual peptide or receptor structures does not necessarily result in the accurate modeling of the interaction (Supplementary Fig. 5).

We assessed the generality of our approach on a large, non-redundant set of 96 complexes that we curated for this purpose (the Large Non-Redundant, LNR, set; see Methods). Modeling of this set reveals that almost 50% of the interactions are modeled within 2.5 Å and about 60% are modeled within 5.0 Å RMSD, a performance slightly better than the non-motif set, but inferior to the motif set, as might be expected (Fig. 1b, upper panel). When calculated overall atoms of peptide interface residues, 37% of the interactions are modeled within 2.5 Å RMSD (Fig. 1b, lower panel).

**Motifs are well modeled and can be identified by high pLDDT**. Given the particularly good performance of AF2 for interactions of proteins with peptides containing a known binding motif (Fig. 1b, blue lines), could we infer the position of a motif based on our predictions? Fig. 2a shows heatmaps reflecting per residue RMSD, together with information about motif residues for the motif set. In most of the complexes, motif residues show considerably lower RMSD values. For some of the peptides in the non-motif set (Fig. 2b), we could identify a similar pattern. For example, we found that the interaction between yeast MAPK Fus3 bound to a peptide derived from MAPKK Ste7 (pdb 2b9h), has a known binding motif[38] that was not annotated in our previous study[13].

Quite a few longer stretches of amino acids are modeled with low RMSD values (Supplementary Fig. 6), providing a good starting point to look for such new motifs. Unfortunately, however, in a real world scenario the peptide structure and the corresponding RMSD values of the models are not known. Luckily, for each model AF2 provides as output also a residue-level confidence estimate, pLDDT (predicted Local Distance Difference Test[39]). Inspection of the corresponding heatmaps shows considerable correlation between the two measures (Fig. 2a, b), as was shown previously for AF2 predictions[28]. A plot of RMSD and pLDDT values for all peptides predicted in this study reveals that this is a general feature: pLDDT values above 0.7 consistently represent accurate predictions within 2.5 Å RMSD, while values below predominantly reflect worse predictions (Fig. 2c; 75% of residues with pLDDT > 0.7 are modeled accurately, while only 8% of the accurate predictions are missed). Average pLDDT>0.7 (calculated over peptide residues) is also predominantly associated with high DockQ[40] values (>0.6) representing medium-to high quality models (This association is stronger than that of normalized Buried Surface Area of models; Supplementary Fig. 7). This suggests that AF2 predictions may be used to reliably identify correct models, and more importantly, previously unidentified motifs.

**AF2 models identify many interface hotspots**. In addition to the identification of the main binding determinants of the peptide, peptide–protein docking aims to provide information about the binding pocket on the receptor. Many receptor interface residues are indeed identified by the AF2 models (Fig. 3a). For the motif set, AF2 provides comparable, although slightly lower recovery of receptor interface residues to PFPD, however for the non-motif-set the recovery rate is significantly lower. Detailed inspection reveals that PFPD can model a less accurate peptide conformation into the correct binding site (see also Fig. 1c, left), resulting in overall better recovery of the binding site, as also reported previously[41]. In turn, AF2 usually generates accurate models once a binding site is identified, but these do not necessarily cover the full site. Still, in most cases AF2 finds at least one residue in the receptor binding site, providing a good starting point for further examination of the predictions using low throughput experiments[6].

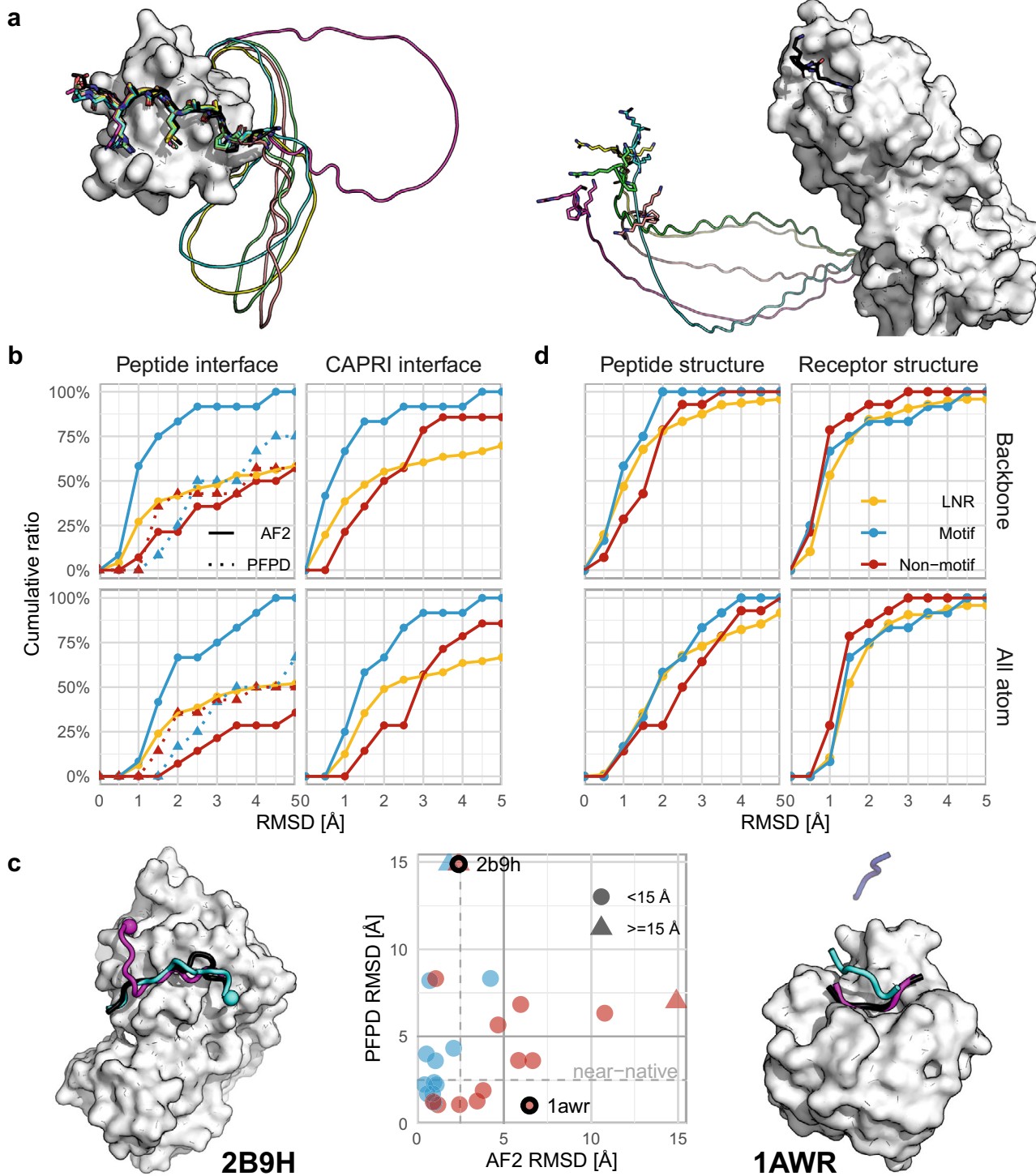

**Fig. 1 AF2 can be adapted to accurately model many peptide–protein interactions. a** Concept of peptide–protein docking with a poly-glycine linker, successfully identified as unstructured region and modeled as a circle, placing the peptide at its correct position (left; example PDB ID 1ssh[63]), or out in space (right; example PDB ID 2orz[64]). The native peptide is shown in black. **b** Cumulative performance of AF2 and PFPD (dashed lines) on motif (12 complexes, blue), non-motif (14 complexes, red), and LNR (96 complexes, AF2 only, yellow) sets, as measured over the interface residues of the peptide (Left - Peptide interface: after aligning the receptor) or the full interface (Right - CAPRI Irms: after aligning the whole interface. See also Supplementary Fig. 2). **c** Correlation between performance of AF2 and PFPD for the motif and non-motif PFPD sets. Triangles indicate values over 15.0 Å RMSD. Left: PDB ID 2b9h[38], interaction between MAPK Fus3 and a peptide derived from MAPKK Ste7, where PFPD positions the peptide within the pocket, but in a flipped orientation (N-termini are indicated by spheres). Right: PDB ID 1awr[65], interaction between CypA and a HIV-1 Gag polyprotein derived peptide, where AF2 models the peptide within the pocket but in a wrong position. Shown are the peptide structures generated by the linker model (cyan, blue) or PFPD (magenta), and the crystal structure (black). **d** Overall assessment of performance measured for the individual partners (after aligning each separately). Source data are provided as a Source Data file.

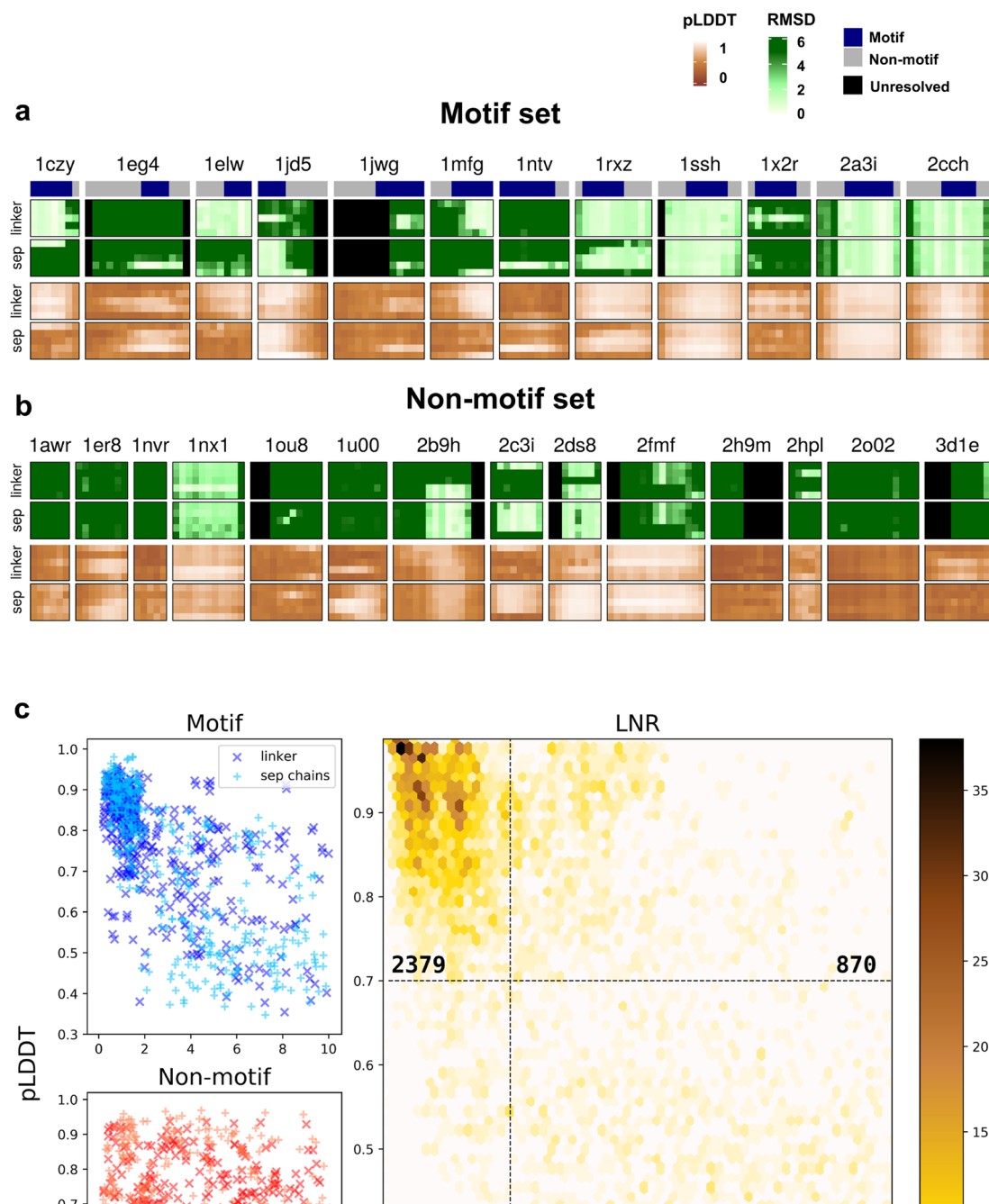

Encouraged by the identification of peptide motifs and the binding pocket residues (Figs. 2a and 3a), we next investigated how well interface hotspots are recovered in AF2 models. For this, we performed computational alanine scanning, both on models and native structures (Fig. 3b), using Rosetta alanine scanning[6]. This simulates a real world scenario where a model would be used for interface hotspot detection, compared to the ground truth

based on the crystal structure (assuming optimal performance of the alanine scanning protocol). Correlation is very strong for accurate models (within 2.5 Å RMSD: Spearman's $\rho = 0.76$ and 0.65, for the LNR set peptide and receptor residues, respectively; all with $p$ values $\ll 10^{-40}$), but also significant overall (corresponding Spearman's $\rho = 0.51$ and 0.34, Fig. 3b and green dots therein). Hotspots are well recapitulated (see Supplementary

**Fig. 2 Accurate modeling results in identification of motif residues and correlates with high pLDDT values. a** Heatmaps of per residue RMSD (shades of green) for peptide–protein interactions with known motifs (motif set), with the top bar highlighting the motif region in dark blue, followed by corresponding pLDDT heatmaps (shades of brown). Each row corresponds to the network parameters used to generate the model (1 through 5). **b** Heatmaps for the non-motif set. Note 2b9h, for which reinspection of the literature reveals a motif. **c** Scatterplots of pLDDT *vs.* RMSD values for the different datasets, representing peptide per residue values of the AF2 models. For the LNR set, clusters of points are represented by hexagonal bins, colored in shades of yellow according to the number of residues in the cluster. The number of observations in each subsection is noted therein. The motif set is shown in shades of blue and the non-motif set in shades of red. Models with linker are represented by dark colored "x" and models with no linker (separate chains) are represented with pale "+". Source data are provided as a Source Data file.

Data 2), featuring only few false positives (i.e., wrong hot-spot predictions; upper left quarter of plots in Fig. 3b), however more false negatives (i.e. missed interface hotspots, lower right quarter of plots therein). Many of them are associated with inaccurate structures, in particular peptides modeled into space (maroon dots on the horizontal 0 value line in Fig. 3b), but also well-modeled structures can miss hotspots identified with the native structure. Thus, while it has been discussed that AF2 is not to be used for the modeling of structural effects of point mutations in the query sequence[42], models can still be the basis for alanine scanning and other structure-based characterizations.

**Peptide sequence plays a crucial role in successful docking**. To better understand AF2 dependency on peptide sequence, we tested an extreme case, in which the whole peptide sequence is replaced by poly-alanine. Performance was dramatically reduced, in particular, when a conserved motif was removed (Fig. 3c). It is noteworthy that a few complexes in each of the datasets were still modeled within 2.5 Å RMSD. But overall, without any information about the peptide sequence, AF2 is not able to successfully model the peptide–protein complex structure.

**AF2 models binding-induced conformational changes**. One of the most challenging tasks in protein docking is the modeling of conformational changes that occur upon binding. Given the success of AF2 in modeling both receptor and peptide conformation (Fig. 1d), we hypothesized that these cases would particularly benefit from this approach that models peptide binding as part of the folding process. Figure 4 shows examples, in which a C-terminal helix positioned in the binding site is removed to make place for the peptide (Fig. 4a), or a beta hairpin loop becomes disordered upon peptide binding (Fig. 4b). In both cases, AF2 correctly predicts the complex structure. When using AF2 to model the free receptor, it also recovered the bound conformation (rather than the free, unbound structure of the receptor). This indicates that the bound conformation was learned, and AF2 predicts the bound conformation by default, even without the presence of the peptide (for these examples). In general, AF2 will tend to clearly favor one conformation. Modeling proteins with multiple conformations has been reported to be a challenging task for AF2, only possible by downsampling the MSA and introducing templates[43].

**What has AF2 learned?** In order to unravel the secrets of AF2 success for peptide–protein docking, we performed additional analyses that shed light on the determining features of the peptide and receptor that make this success possible. Can the high performance be attributed mainly to memorization, or has it actually learned basic features?

On the peptide side, we showed that peptide sequence is crucial for successful modeling of interactions to a receptor (Fig. 3c). Additional peptide features that could affect the quality of AF2 peptide-receptor models include the peptide length and secondary structure. Peptide length seems to have little effect on

AF2 success (Fig. 5a, Supplementary Fig. 8a), as has already been shown[35]. This is in contrast to its effect observed on peptide docking with PFPD[13]. Regarding secondary structure, helical peptides are particularly well modeled (Fig. 5b, Supplementary Fig. 8b). This indicates that AF2 is biased towards helical structures, as was reported for other NNs[44], possibly due to its over-representation in the learning set.

Could AF2 have copied the peptide–protein complex structures from templates in the training set? Although the training set and learning protocol of AF2 were carried out on single chains[28], there may be cases in which the peptide complex structure was nonetheless learned as part of the chain, as in the case of a synthetic fusion of segments, uncleaved pro-proteins, or single chains with residues located within the binding site (e.g., their own tails). We found only 13 possible such examples in the LNR set (see Methods), among them five with precise recapitulation of the interaction (*i.e.*, 5% of the LNR set). For these five, highly accurate models were generated (within 1.5 Å RMSD, see Supplementary Data 3). Success in these cases could be a result of the direct memorization of precisely those structures. For the remaining eight, half are successfully modeled, reflecting performance similar to the overall set (see Fig. 5c for specific examples). We conclude that at most a few cases of successful modeling could result from direct memorization. In fact, we note that even if the solved structure is provided as an additional input (for the trained NNs model1 and model2; see Methods), it is rarely used, and only in a few cases does this improve modeling (Supplementary Data 4).

We used the same approach to model an additional set of peptide–protein interactions that we had removed from our LNR set, due to the context in the solved structure that could prohibit accurate modeling of the complex. This includes post-translational modifications (PTMs) of peptides or receptor interface residues, additional ligands at the interface that contribute to peptide binding, or crystal contacts that significantly affect the peptide conformation (see Methods). This type of information was not directly included in the training or inference pipeline of AF2, and is not provided as input. Surprisingly, modeling performance for interactions that include PTMs or a ligand at the interface is comparable to that of the LNR set (Fig. 5d). AF2 succeeds in modeling over 35% of these complexes within 2.5 Å, despite training only on single chains, canonical amino acids and without bound ligands. We believe this could be attributed to learning the structures as they occur in the PDB database, emphasizing that while AF2 may be optimal for structural modeling, it lacks a more intricate understanding of the details of biophysics underlying some of the peptide–protein interactions.

To summarize, we show here that AF2 can model not only monomer structures but also many of the interactions between peptides and protein receptors. This is true in particular when a peptide binding motif is available, and even in challenging cases where the monomer changes its conformation upon peptide binding. We also highlight some limits of AF2, and details not learned that need to be completed using complementary approaches.

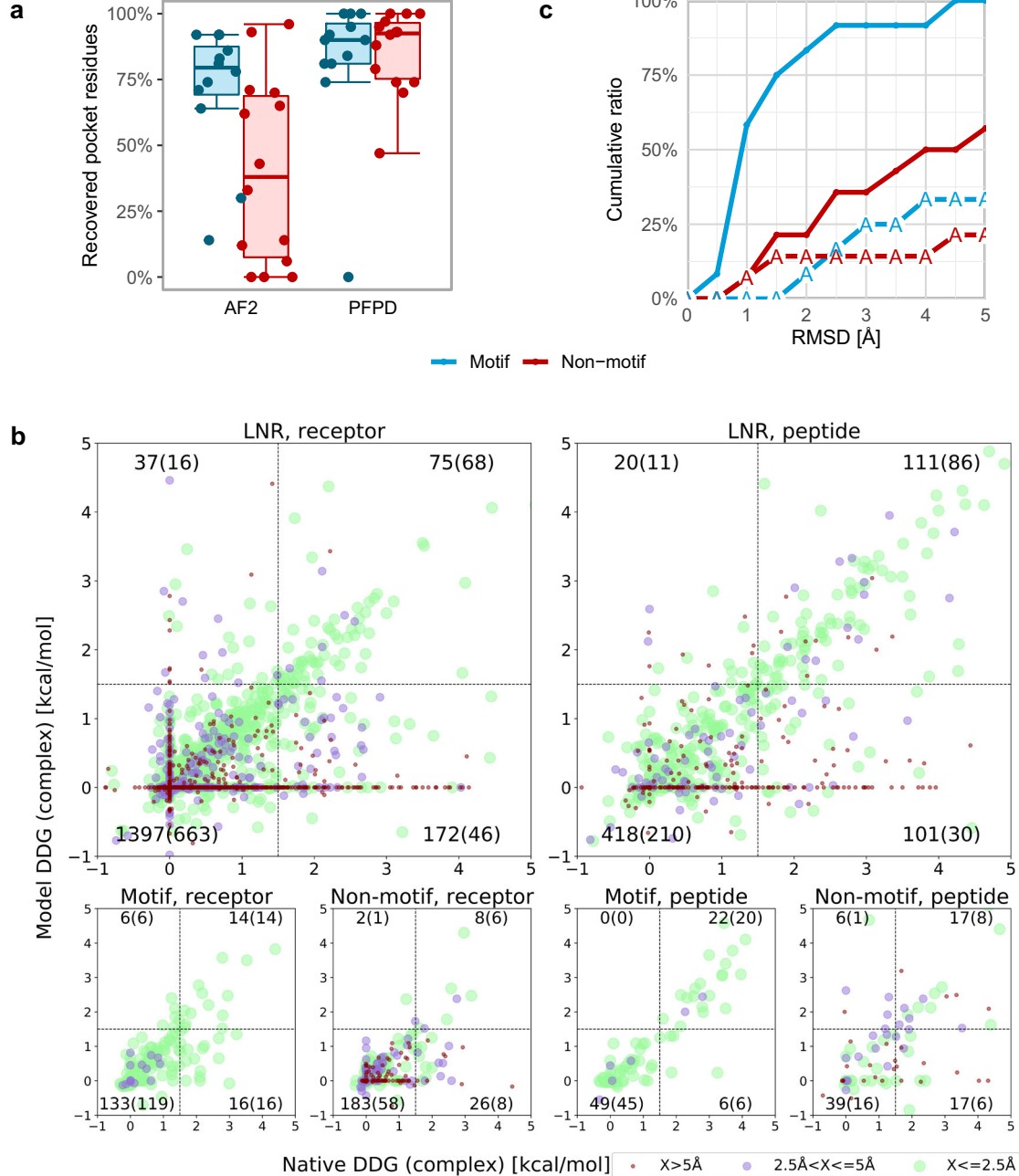

**Fig. 3 Recovery of peptide–protein interface characteristics by AF2 models. a** AF2 models identify a significant fraction of the receptor binding pocket residues, even though coverage is reduced compared to PFPD models. Box-and-whisker plots are shown for motif ($n = 12$, blue) and non-motif ($n = 14$, red) sets, with median as center line, quartiles as box limits, and lower/upper whiskers extending to the maximum/minimum data points within the interquartile range, respectively. **b** Computational alanine scanning recovers well-predicted interface hotspots, mostly for accurate models (within 2.5 Å RMSD): Comparison of computational Rosetta alanine scanning results applied to the best model vs. the native structure. The vertical and horizontal lines represent the threshold $\Delta\Delta G = 1.5$ kcal/mol used to define interface hotspots. The different colors represent distinct bins of RMSD (of the model used for alanine scanning). The number of residues in each quadrant is indicated (counts from models below 2.5 Å are shown in parentheses). **c** Performance is significantly reduced if the peptide sequence is changed to poly-alanine, both for motif and non-motif sets. Source data are provided as a Source Data file.

## Discussion

In this study we have applied the AF2 protein structure prediction protocol to predict peptide–protein complex structures. Without any further training, and only minor optimization of the runtime parameters, we were able to reach an accuracy comparable to that of state-of-the-art protocol specifically developed for the task of peptide docking (Fig. 1).

AF2 has many advantages: it is much faster than established protocols such as PFPD (around 20 min for five models––when using

the MMSeqs2 server[45] for MSA generation, which is the bottleneck of the protocol–vs. a couple of hours for docking with PFPD), with no significant trade-off between model quality and runtime. An additional advantage is that AF2 only requires sequences as inputs; no structural information is needed. Finally, for AF2 predictions, clear failures are often easily identified as structures in which the peptide does not interact with the receptor, but rather points out into space.

AF2 has also disadvantages: The diversity of interfaces is usually low, in line with observations that such models quickly

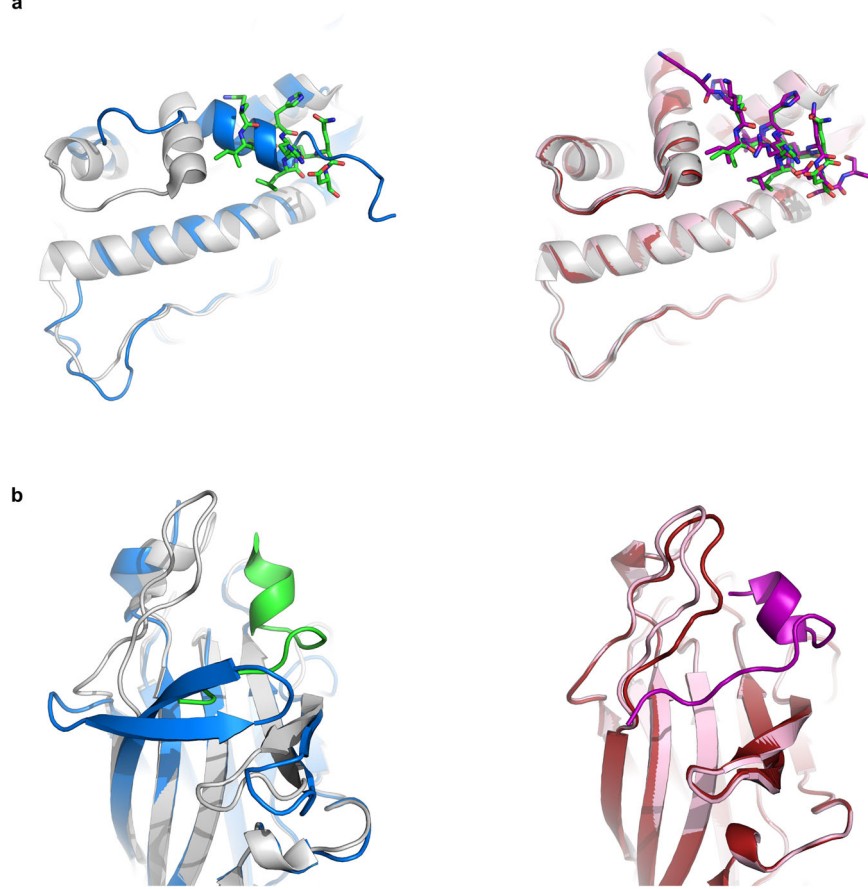

**Fig. 4 AF2 modeling of interactions involving significant changes in receptor conformation. a** Estrogen receptor alpha - peptide derived from nuclear receptor cofactor 2 (PDB ID 2b1z[66]). Left: In the free receptor conformation (PDB ID 3ert[67]) the C-terminal helix occupies the peptide binding site. Shown are the solved crystal structures of the unbound (blue) and bound conformation (receptor in white, peptide in green). Right: AF2 models the correct receptor conformation, and positions the peptide accurately in the binding site (RMSD < 1.0 Å). Shown are the AF2 model (unbound receptor: maroon, bound receptor: pale pink, peptide: magenta), compared to bound conformation (white). **b** EphB4 receptor––ephrin-B2 antagonist (PDB ID 2bba[68]). Left: In the free receptor conformation (PDB ID 3etp[69]) the J–K loop forms a β-hairpin. Right: Upon peptide binding, this hairpin becomes disordered and assumes a flexible loop conformation, which is recovered in the AF2 model. Coloring as in **a**.

converge on a minimum[46]. While this characteristic is often advantageous for reducing false positives, it does not allow for wide sampling of conformational space and assessment of the energy landscape (as is possible with other protocols, e.g., PFPD), even though this could be addressed by increasing the seed number (which did not contribute to improved performance in the present study, see Supplementary Fig. 4).

AF2 can predict peptide–protein complexes even though it was only trained on monomer chains. Could the success of AF2 peptide docking still be due to some memorization of interfaces? Our results suggest that this is not the case. First of all, even when the monomer structures are modeled at high precision (Fig. 1d), this does not necessarily guarantee high-resolution models of the interaction (Supplementary Fig. 3). Moreover, only very few monomer structures are available that accurately cover the interface and could serve for memorization (5% of the LNR set, Fig. 5c and Supplementary Data 3), and even when the crystal structure is provided as input, it is not necessarily used, or helpful (Supplementary Data 4). Still, AF2 succeeds in peptide docking, indicating that the underlying principles for peptide–protein interactions were well captured and learned - again supporting the view of peptide–protein docking as a protein folding problem.

The ultimate way to assess memorization of existing structures is to assess performance only on structures not included in the training set of AF2 (i.e. structures published after 4/2018).

Reassuringly, models of this subset (10/96 structures) are modeled at similar, or even better precision: six out of these complexes are modeled within 2.5 Å, four of these even within an impressive 1.0 Å RMSD. However, this set includes only one interaction involving a new ECOD domain. While AF2 failed for this complex, no general conclusion can be made based on one example. However, the important measures of success are the recapitulation of the interface between the peptide and receptor, and AF2 was not trained on that.

To conclude, although remarkable for a method that was not trained for the task, the performance of AF2 is not good enough to assume some hidden overfitting during the training process that we are not aware of. On the other hand, our analysis of complexes harboring PTMs or bound ligands also resulted in a similar performance which indicates that memorization is indeed present in the network. This also points us to challenges ahead that will need to be addressed to further improve peptide docking using AF2.

A significant advantage of this protocol lies in its potential to also model considerable conformational changes of the receptor upon binding. This is due to folding both the receptor and the peptide simultaneously. This would be of special importance in cases where binding induces conformational changes to the receptor (Fig. 4). This is also an advantage over template-based methods - AF2 can dock peptides to proteins for which close

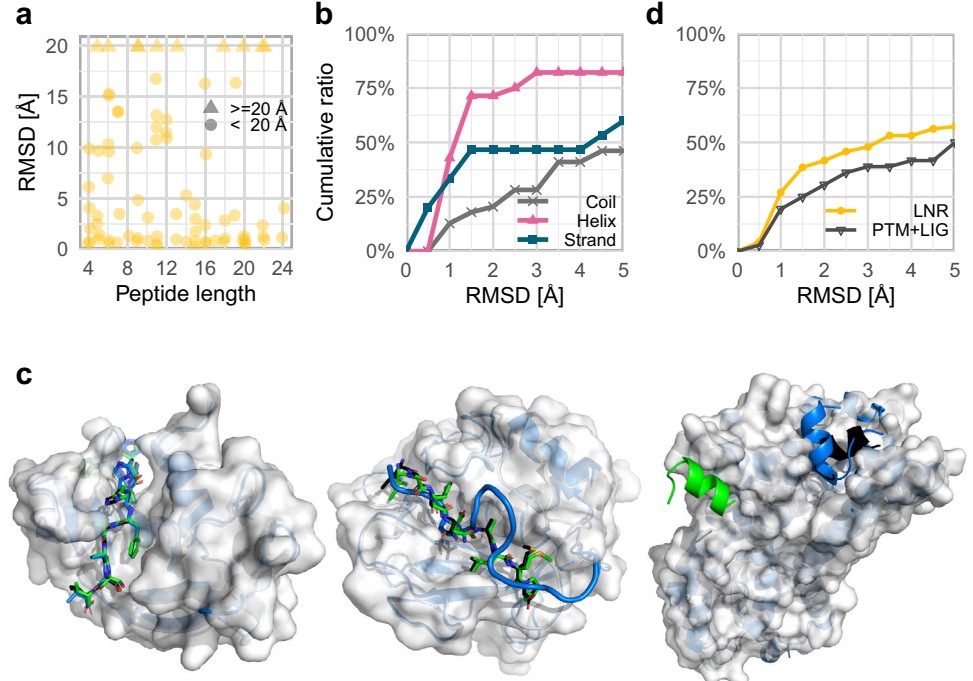

**Fig. 5 Analysis of potential factors influencing AF2 peptide docking performance. a** Peptide length does not affect docking performance: successful models are generated for long as well as short peptides (See Supplementary Fig. 8A for the distribution of peptide lengths in the LNR dataset). Values over 20 Å are indicated by triangles. **b** Alpha helical peptides are modeled particularly well. Cumulative plot of performance according to the secondary structure of the peptide. (See Supplementary Fig. 8b for the corresponding distribution of secondary structures). Helical, beta strand, and coiled peptides are colored in pink, blue and gray, respectively. **c** Examples of potential memorization by AF2 from similar monomer structures: Left: Highly accurate prediction where a precise coverage of the structure of the peptide-receptor interface is available (native PDB ID 1ssc[70] and memorized PDB ID 1a5p[71]). Center: Inaccurate template but successful modeling (native PDB ID 3ayu[72] and memorized PDB ID 5ue4[73]). Right: Inaccurate template and failed model (native PDB ID 2x72[74] and memorized PDB ID 5dgy[75]). In the latter two cases the model is most probably not built based on the inaccurate template. For all panels, the native receptor is shown in white, and native peptide in black, the memorized peptide in blue, and the modeled peptide in green. **d** Cumulative plot of performance on a set of peptide-receptor complex structures that were excluded due to post-translational modifications or bound ligands that influence the peptide structure: AF2 modeling is not dramatically impaired. The LNR set is shown in yellow, and the PTM + LIG set is shown in black. Source data are provided as a Source Data file.

homologues are not available in the PDB. It is yet to be assessed whether AF2 can dock a peptide to a receptor if only the unbound conformation of the complex was available during training.

A surprising feature is the ability of AF2 to model peptide–protein complex structures without available MSA information for the peptide. This is particularly surprising since the cornerstone for accurate AF2 predictions is learning residue conservation and co-evolution through contextual processing of MSAs, and benchmarked performance was shown to drop significantly with a decrease in the number of effective alignments for a query[28]. The impressive success for peptide docking, albeit completely lacking MSA coverage for the peptide side (in the context of the complex as modeled in this study), is non-trivial. This is yet another indication that the essence of peptide binding can be implicitly captured as an extension of folding.

Short linear motifs play an important part in binding partner and substrate recognition between proteins[47]. For most interactions in the *motif set*, per residue RMSD, and more importantly for prediction, pLDDT values, correctly identify the motif residues within a peptide (Fig. 2). This is important, since in high-throughput experiments such as beads or phage display[48,49], a longer stretch of binding peptides is detected, without information about the exact location of the motif within, and often without information about their binding site on the receptor structure. Using AF2 for docking these peptides can be a rapid way to process these results and identify previously unknown motif instances together with their probable binding site and

conformation, e.g., using pLDDT for motif identification and computational alanine scanning for characterization of the receptor binding site. In turn, for peptides without resolved binding motifs, for which AF2 currently does not perform as well, it might be interesting to investigate how inclusion of local MSA information extracted from the MSA of the full source protein could impact the overall accuracy of the complex.

We have presented here a straightforward adjustment of AF2 for peptide docking. Further fine-tuning will without a doubt improve the protocol and expose new features that contribute to successful modeling. Parameters to calibrate include more sophisticated approaches to MSA generation which might result in improved docking and better motif detection, as indicated previously[34]. In addition, the very recent publication of AlphaFold-Multimer may supply another avenue for peptide–protein docking[50]. Finally, the partial orthogonality of performance of AF2 and PFPD (Fig. 1c) bears promise for improved peptide–protein docking by combining these approaches.

To summarize, on the conceptual side, the fact that AF2 was trained and tested on monomeric structures, but can be successfully applied to model peptide–protein interactions, reinforces our view of peptide-receptor binding as complementation of the final structure of a monomer. On the practical side, the experiments reported here and elsewhere pave the way towards exciting avenues for peptide–protein docking and the study of peptide-mediated interactions in general. We believe that by using such approaches, many of the long existing obstacles of the field could

be overcome, allowing the study of many more biological systems at high structural resolution.

## Methods

**Structural modeling with AF2**. Modeling was performed using the publicly available AF2 repository[28], with each of the five trained model parameters. The input included the query sequence and MSA from MMseqs2[51], without using any templates (unless otherwise noted). No additional refinement was performed on the models.

Both MSA generation and AF2 predictions were run using the code of ColabFold, a publicly available Jupyter notebook[30], slightly modified for local batch runs, on a local GPU cluster. The modifications did not affect running parameters, only the mode of providing input data was changed.

**Sequence collection and formatting**. The sequences of the receptor and peptide to be modeled were extracted from the SEQRES lines of the PDB files, to account for the expressed construct rather than the structure resolved in the PDB. Unknown residues and terminal modifications were removed. Then, the sequences of each pair were concatenated with a linker of 30 glycine residues (for the runs containing linkers), in the order of (N-terminus)receptor-linker-peptide(C-terminus).

**Optimization of running parameters**. We inspected the contribution of several input parameters to the performance of AF2. Four factors were evaluated (the first option stated was our starting default): poly-Gly linker or separate chains, number of recycles (three or nine), the use of environmental sequences (yes/no), drop-out (no/yes), and the number of seeds (one or five). For every complex in the motif and non-motif sets, $2*2*2*2*5*5 = 400$ models were generated.

**Modeling of Poly-A-receptor interactions**. Poly-A peptide docking was carried out on the motif and non-motif datasets, by mutating the peptide residues to Alanine (in the query sequence), keeping the original peptide length for each structure. The models were then generated as described above.

**Template-based modeling**. As AF2 can read in templates based on HHSearch results, we created an alignment file for every complex, providing the receptor and peptide chains as separate hits, with perfectly matching alignments. By providing the path of the directory containing all the mmCIF files for these structures, AF2 was able to parse and process these templates. Since out of the five trained models, only model 1 and 2 can use templates, these predictions were only run using these models, with the previously selected configuration (recycles = 9, no drop-out, with environmental sequences, one seed, both with poly-Gly linkers and separate chains).

### Structure datasets compiled and used in this study

*Generating a comprehensive set of peptide–protein complexes.* For robust assessment of a modeling protocol, it is important to generate a non-biased, non-redundant dataset. For ease of curation and initial analysis, the PDB was queried for entries with two chains only, and filtered for those having possible protein-peptide interactions according to the following criteria: 1. One chain must be over 30 amino acids long, and one chain must contain between four and 25 amino acids (with at least three amino acids resolved in the solved structure). 2. The peptide chain must have at least two residues within 4 Å distance from the protein chain. This yielded a total of 16,931 structures belonging to 1102 ECOD domains[52]. Once possible interactions were identified, the following filters were applied: 1. Remove structures with peptide residues annotated as UNK, 2. PDB-range and seq-range fields must agree on the indices of the receptor domain according to ECOD annotation, 3. Apply symmetry operations (from the PDB entry) on the asymmetric unit and check for possible crystal contacts that may affect the bound conformation of the peptide. Remove cases where at least 20% of the peptide residues are in contact with symmetry mates. Structures from ECOD families represented in the motif and non-motif sets were removed. The resulting list was manually validated, and structures were set aside that contain a peptide conformation that might be influenced by context not included in the input (e.g., structures containing ligands in the vicinity of the peptide binding site or peptides with modified residues, such as PTMs). The final list after filtering and manual validation consists of 96 peptide–protein complexes (Large, Non-Redundant: LNR set), and 13 interactions involving PTMs or bound ligands (PTM + LIG set, evaluated separately in Fig. 5d). See Supplementary Data 1 for the full datasets.

*Identifying monomers resembling peptide–protein interaction.* Monomer chains that could have been used to memorize peptide–protein interactions were detected by employing two orthogonal approaches: (1) Based on UniProt annotations: We extracted UniProt chain annotations from the SIFTS database[53] for all the members of each ECOD family with a representative structure in the dataset, and examined structures with more than 1 UniProt annotation per single chain. (2) Based on structural analysis: all members of the relevant ECOD family were

superimposed, and occupancy of the pocket corresponding to the peptide binding site by the receptor monomer was detected. For both approaches, a list of candidates was assembled and manually filtered to verify mimicking interactions.

**Comparison to PIPER-FlexPepDock**. Complexes of the motif and non-motif sets were modeled using PFPD with default settings, as was benchmarked[13]. For each complex, top 10 cluster representatives by FlexPepDock reweighted score were selected for comparison with AF2. Note, that for this assessment, we used the PFPD set (26 complexes) consisting of two subsets, one with and one without reported motifs (motif and non-motif sets), as described therein.

### Analysis of models

*RMSD calculations.* Backbone and all-atom RMSDs of the peptide interface residues (rmsBB_if, rmsALL_if) and the whole interface (rmsBB_allIF, rmsALL_allIF) were calculated using Rosetta FlexPepDock (release 2020.28[16]), after aligning the receptor (the interface is defined as Cβ atoms within 8.0 Å distance across the interface). We also report the slightly different CAPRI interface metrics: (Irms and Lrms)[54], which are calculated over both peptide and receptor interface residues, after aligning the said residues of the native and model structures (see Supplementary Fig. 2). RMSD values for the individual peptide and receptor structure were calculated using PyMOL python API (v2.2.0), using the align command, without any cycles and rejection of atoms.

The following command was used for rescoring models and calculating RMSD values:

```
> FlexPepDocking.linuxgccrelease -native ${complex}
_native.pdb \ -flexpep_score_only -out:file:score_only
${complex}.score.sc \ -s ${complex}*_models.pdb
```

*By-residue RMSD calculations.* Model complexes (protein-peptide) were aligned to the native complex as described in the previous paragraph. All-atom RMSD was computed using BioPandas python module[55] for each peptide residue pair (model-native), skipping residues that were unresolved in the native structure. Atoms lacking in the models (such as OXT) were also ignored.

*By-residue LDDT predictions.* We extracted the per residue LDDT prediction values from the b-factor column of the structural models output by AF2[28].

*Binding pocket calculations.* Binding pockets on the receptor were defined as those residues that have at least one backbone atom located within 8.0 Å to a peptide backbone atom. The calculations were performed with a PyMOL script[56].

*Computational alanine scanning.* Alanine scanning was performed using the Robetta alanine scanning implementation[6].

*DockQ and buried surface area calculation.* The DockQ model quality metric was computed with the default settings and parameters, using a two-chain configuration (receptor: A, peptide: B)[40].

Buried surface area was computed using the Rosetta Interface Analyzer[57] in default settings, with no additional configurations. The metric presented in Supplementary Fig. 7 is "dSASA_int" (solvent accessible area buried at the interface, in square Ångstroms) normalized for each pdb to the maximal value of its models.

*Visualization.* Visualizations were performed with custom R and Python scripts, using packages ComplexHeatmap[58], ggplot2[59], matplotlib[60], and PupillometryR[61]. To visualize structures, we used PyMOL[56].

**Reporting summary**. Further information on research design is available in the Nature Research Reporting Summary linked to this article.

## Data availability

All source data are provided with this paper. These data, as well as models generated in this study, are available at https://github.com/Furman-Lab/Peptide_docking_with_AF2_and_RosettAfold[62]. PDB entries used in this study and their corresponding hyperlinks are listed in Supplementary Data 5. Source data are provided with this paper.

## Code availability

The code for processing, analyzing and visualizing the results is available at: https://github.com/Furman-Lab/Peptide_docking_with_AF2_and_RosettAfold[62].

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

## Acknowledgements

We are grateful to Sergey Ovchinnikov, Martin Steinegger and Milot Mirdita, and anyone else that has helped provide notebooks to run AF2. This work was supported, in whole or in part, by the Israel Science Foundation, founded by the Israel Academy of Science and Humanities (grant numbers 717/2017 and 301/2021 to O.S.-F.) and the US-Israel Binational Science Foundation (grant number 2015207). J.K.V. is supported by a Marie Sklodowska-Curie European Training Network Grant #860517.

## Author contributions

T.T. conceived the idea for the research and performed initial experiments, T.T., J.K.V, O.A., and O.S-F. refined the concept and realized the final implementation. J.K.V. adapted the existing code for high-throughput local running, and T.T., J.K.V, and O.A. performed the experiments, T.T. and J.K.V processed the raw results, T.T., J.K.V, O.A., and O.S.-F. analyzed the results and wrote the manuscript. Z.B.A and A.K generated the dataset, analyzed the results, and contributed to the manuscript, O.S.-F. supervised the project and acquired funding.

## Competing interests

The authors declare no competing interests.

## Additional information

**Peer review information** :Nature Communications thanks Arne Elofsson and the other, anonymous, reviewer for their contribution to the peer review of this work. Peer reviewer reports are available.

