## [Peer Review File · Nature Communications]

REVIEWER COMMENTS

Reviewer #1 (Remarks to the Author):

This manuscript describes and evaluate a protein-peptide modelling approach making use of the recently released AlphaFold2 code. The idea is similar to another study posted in BioRxiv (citation 40 in the manuscript): Adding a linker between the protein and the peptide. The method is demonstrated on a set of 162 non-redundant protein peptide complexes and compared to the previously published protocol by the same group and co-workers (PIPER-FlexPepDock (PFPD)). The overall performance is impressive considering AF2 was not trained on complexes. Overall the results are clearly presented with thorough analysis and comparisons between approaches.

I only have rather minor comments.

- 1) When discussing the RMSD values shown in Figure 1A), the authors should indicate how the superimposition was done (probably on the protein alone). Also I would suggest to indicate in the figure as vertical lines the CAPRI criteria (I-RMS limits).
- 2) The authors remark that the two approaches (PDPD and AF2) show no correlation in performance, which opens the way to a combination of the two. In a real scenario however one will not know which one might give the best results. How would you then combine the results / select one or the other method? Some discussion about this should be added.
- 3) Related to 2, it would be interesting to show if a Rosetta refinement of the AF2 models would lead to clear binding funnels. Especially for case where one of the two approaches fails. Could the presence or absence of a funnel point to the correct model?
- 4) Some more info on the benchmark could be provided, especially the distribution of sequence lengths and also the fraction of extended vs helical conformations.
- 5) Related to 4), all data, notebooks, analysis scripts and generated models should be made freely available

6) I would suggest to do a careful reading of the manuscript to improve readability, e.g. add here and there some pronouns (it does currently feel a bit as if it has been very quickly written...)

Reviewer #2 (Remarks to the Author):

In this manuscript, the authors examine the possibility to use AlphaFold2 to dock peptides. The results are quite impressive (as expected from

First, it has to be mentioned that the authors seem to have rushed to submit the paper. They have used work by the Colab alphafold team (Ovchinnikov and Steinegger mainly I think) and run a preliminary version of their notebook on a previously published dataset. They did not even bother to read the code from DeepMind and realize that it is easy to perform this type of docking without using the Gly-linker. Obviously, the study should be replaced without the GLY-linker, but I do not expect the results to change significantly. I will assume that this is the case for the rest of my review.

As has been reported in numerous Twitter posts AF2 can be used in many different ways beyond the folding of a single domain. Here, the authors show that it can dock and fold approximately 25% of the protein-peptide complexes. They compare this with state-of-the-art methods and show that the performance is better than when the motif is not used, but worse than when it is known. They also use the docked peptides to try to extract motifs.

Although I think the paper is interesting, I can not help to not feel a bit disappointed - the paper does not really provide any information on (1) the best strategies to perform the docking (2) the limits (why does it not work in some cases, (3) how AF2 can achieve this impressive performance (it certainly learns something about folding as the authors say - but what and how, (4) any novel biology, or (5) how we can achieve even better performance.

To address these questions I would suggest that the authors do (in addition to rerun everything without the linker)

1. Examine different strategies for docking (one obvious test is to see what happens if you run several recycles, or generate several models. Is it just a bit of randomness when it works or not or if it really is something that separates the failed and successful cases.

2. Examine what happens if the template (i.e. the structure) of the protein is used as input (and also the same for the peptide). How does that change things ?

3. Examine what happens when you have peptides that should not bind (or try to do an Alanine scanning of the peptide). Does that matter etc ? What happens with a poly-A ?

4. Run this on a much larger dataset of peptide-proteins with unknown structures (the SLIM database). This might provide some really interesting biological findings.

Yours

Arne

We thank the two reviewers for their quick, helpful and supportive comments that have allowed us to significantly improve our manuscript. While our initial submission was quickly performed and written, we have taken the time and effort to significantly revisit and extend our study and are now happy to report a comprehensive and much improved manuscript. Overall, the baseline result has not changed: AF2 can accurately predict a large fraction of peptide-protein complexes, without dedicated training on peptide-protein docking. Our analyses suggest that memorization does not play a major role and support a more general notion that AF2 applies what it has learned from monomer structures to modeling of peptide-protein interfaces, reinforcing the view of peptide protein binding as complementation of protein folding.

We performed initial calibration of optimal AF2 performance on the motif and non-motif sets previously assessed (total of 26 complexes). Calibration included the investigation of several parameters: (1) prediction by providing the peptide as separate chain, compared to prediction using a poly-glycine linker; (2) increase the number of recycles; (3) use of environmental sequences; (4) use of stochastic dropout; and (5) increase the number of random seeds. Based on these runs, we decided to proceed with a protocol that (1) combines both submissions with separate and poly-glycine linked peptide chains; (2) uses 9 instead of 3 recycles; and (3) inclusion of environmental sequences (Supplementary Figure 4).

Next, we invested much time and effort to curate the validation benchmark. We now present a greatly improved, curated dataset of almost 100 complexes on which we benchmarked AF2 performance (Supplementary Table 1). All entries in our datasets are non-redundant at the ECOD domain level, ensuring unbiased representation of peptide-protein interactions. Finally, we inspected the use of models for further analysis of an interaction, and investigated a number of factors that could contribute to the success of AF2 peptide docking reported here. This includes investigation of the importance of the peptide sequence on the one hand, and the availability of templates used in the training or prediction step, to prediction accuracy. These analyses reinforce our notion that peptide-protein interaction specific memorization does not play an important role. Instead, it seems that AF2 is able to apply concepts learned from monomer prediction to the modeling of peptide-receptor interfaces.

Below we include detailed responses to all the Reviewer's concerns and suggestions. We hope that after this detailed and comprehensive review, improvement and extension of the manuscript, we have addressed all concerns and the manuscript can be accepted for publication.

Reviewer #1:

This manuscript describes and evaluates a protein-peptide modelling approach making use of the recently released AlphaFold2 code. The idea is similar to another study posted in BioRxiv (citation 40 in the manuscript): Adding a linker between the protein and the peptide. The method is demonstrated on a set of 162 non-redundant protein peptide complexes and compared to the previously published protocol by the same group and co-workers (PIPER-FlexPepDock (PFPPD)). The overall performance is impressive considering AF2 was not

trained on complexes. Overall the results are clearly presented with thorough analysis and comparisons between approaches.

I only have rather minor comments.

1) When discussing the RMSD values shown in Figure 1A), the authors should indicate how the superimposition was done (probably on the protein alone). Also I would suggest to indicate in the figure as vertical lines the CAPRI criteria (I-RMS limits).

A:

Our measure differs slightly from the standard CAPRI measures: Throughout the paper we report RMSD values calculated over the peptide interface residues, after aligning the receptor. This measure is more stringent, but reflects in our opinion best how well AF2 docks the peptide, as it focuses on the peptide residues, which are usually less well modeled compared to the receptor (and will fail in cases where the receptor structure is wrong).

In **Figure 1B** we also report results for the standard CAPRI I-RMS measure, revealing a similar trend. **Supplementary Figure S2** includes CAPRI I-RMS and L-RMS cumulative plots, where we have added lines indicating CAPRI thresholds for acceptable, medium and high quality models. This is now detailed in the Methods section.

2) The authors remark that the two approaches (PFPD and AF2) show no correlation in performance, which opens the way to a combination of the two. In a real scenario however one will not know which one might give the best results. How would you then combine the results / select one or the other method? Some discussion about this should be added.

A: We agree with the reviewers comment and now relate to this more in the Discussion.

One possibility to identify successful predictions is to inspect pLDDT values (for AF2 predictions), as we show in Figure 2. But this is only the beginning. The optimization of a protocol that combines AF2 predictions with Rosetta FlexPepDock and other approaches is the focus of our currently ongoing work.

3) Related to 2, it would be interesting to show if a Rosetta refinement of the AF2 models would lead to clear binding funnels. Especially for cases where one of the two approaches fails. Could the presence or absence of a funnel point to the correct model?

A: As mentioned above, this is the matter of current in depth exploration, which we plan to report later, and goes beyond the focus of the present study.

4) Some more info on the benchmark could be provided, especially the distribution of sequence lengths and also the fraction of extended vs helical conformations.

A: For the revision, we have put great effort into generating a clean, manually curated dataset for validation (containing 96 complexes), and now provide a full report about this benchmark (termed *Large Non-Redundant set, LNR*), including a detailed list of the different datasets analyzed (**Supplementary Table 1**).

We have included distribution of peptide length (**Supplementary Figure 7A**), alongside an analysis of AF2 performance dependency on peptide length (**Figure 5A**). Similarly, we report

the distribution of secondary structure (**Supplementary Figure 7B**) as well as corresponding performance, highlighting best performance for helical peptides (**Figure 5B**).

5) Related to 4), all data, notebooks, analysis scripts and generated models should be made freely available

A: A table containing the datasets and all the relevant information is now included as **Supplementary Table 1**. All relevant scripts and notebooks, as well as the models are available at a github repository at:

https://github.com/Furman-Lab/Peptide_docking_with_AF2_and_RosettaFold

and linked in the Methods section.

6) I would suggest to do a careful reading of the manuscript to improve readability, e.g. add here and there some pronouns (it does currently feel a bit as if it has been very quickly written...)

A: We thank the reviewer for this important notion. Together with the additional analyses provided in the resubmission, we have made efforts to improve readability and flow of the manuscript. We hope it stands up to the expectations.

Reviewer #2:

In this manuscript, the authors examine the possibility to use AlphaFold2 to dock peptides. The results are quite impressive (as expected from

First, it has to be mentioned that the authors seem to have rushed to submit the paper. They have used work by the Colab alphafold team (Ovchinnikov and Steinegger mainly I think) and run a preliminary version of their notebook on a previously published dataset. They did not even bother to read the code from DeepMind and realize that it is easy to perform this type of docking without using the Gly-linker.

Obviously, the study should be replaced without the GLY-linker, but I do not expect the results to change significantly. I will assume that this is the case for the rest of my review.

A: We examined the results of modeling using linkers and separate chains, as suggested. We found that these two implementations are rather complementary (see **Supplementary Figure 3**). Thus, we decided to incorporate both in the generation of models (*i.e.*, 10 models in total, which corresponds to the number of models evaluated in CAPRI). We select the top model (by RMSD) for analysis.

As has been reported in numerous Twitter posts AF2 can be used in many different ways beyond the folding of a single domain. Here, the authors show that it can dock and fold approximately 25% of the protein-peptide complexes. They compare this with state-of-the-art methods and show that the performance is better than when the motif is not used, but worse than when it is known. They also use the docked peptides to try to extract motifs.

Although I think the paper is interesting, I can not help to not feel a bit disappointed - the paper does not really provide any information on (1) the best strategies to perform the docking (2) the limits (why does it not work in some cases, (3) how AF2 can

achieve this impressive performance (it certainly learns something about folding as the authors say - but what and how, (4) any novel biology, or (5) how we can achieve even better performance.

A: We would like to thank the reviewer for thorough reading of the manuscript and all the points raised.

We believe the reviewer will find the revised version of the manuscript extensive and insightful. We addressed all reviewers' concerns as detailed below. Briefly, we examined several technical modifications and their effect on performance, such as the use of linker and separate chains, network parameters and others. We describe the limitations of the method with regards to recovering interface residues, detected and missed hotspots, and false-positive modeling of, e.g., PTM peptides. We thoroughly examine what AF2 has learned, and demonstrate its relationship with different data biases such as possible “memorization templates”, and other biological characteristics such as the amino acid sequence, secondary structure and peptide length.

To address these questions I would suggest that the authors do (in addition to rerun everything without the linker)

A: We have rerun everything without the linker, and as described above, now include both implementations, as they show complementary performance.

1. Examine different strategies for docking (one obvious test is to see what happens if you run several recycles, or generate several models. Is it just a bit randomness when it works or not or if it really something that separates the failed and successful cases.

A: Additionally to the hyperparameters used in the previous version, we introduced several new widely discussed ones. On the motif and non-motif sets that we used for calibration we evaluated the effects of: (1) polyG linker versus separate chains; (2) number of recycles (0, 3 and 9); (3) number of seeds (1 or 5); (4) stochastic dropout (used or not used); and (5) inclusion of environmental sequence databases.

We report the results for this calibration in **Supplementary Figure 4**. Based on these results, we conclude that the effects of dropout and varying seeds are minor. A large number of recycles is beneficial, and so is the inclusion of environmental databases, to a certain extent. Linker and separate chains work differently, some of the complexes benefitting from one strategy and others from the second. Hence, for the rest of the paper and the larger dataset, we model everything using both linker and separate chains, with one seed, 9 recycles, including environmental databases and no random dropout. We examine the differences between models with and without linker in several analyses (such as hotspot recovery in **Figure 2**), and otherwise present the best model (by RMSD) out of the 10 (5 linker + 5 separate chains) models for each complex.

2. Examine what happens if the template (i.e. the structure) of the protein is used as input (and also the same for the peptide). How does that change things ?

A: With respect to “assisting” AF2 with templates, we investigated 2 distinct aspects:

1. We describe a new results chapter discussing what AF2 has learned. Specifically, we were interested to check if AF2 peptide-docking successes could be attributed to some “memorization”. In such a case, a single chain protein structure recapitulates

the peptide protein interaction (naturally with a tail from the monomer or synthetically with a peptide “fused” to the monomer biochemically). We map such “memorization templates”, and show that such templates are available only for a few peptide-protein complexes in our dataset. The performance for complexes with memorization templates is indeed excellent for the 5 complexes (out of total 96 complexes in the *LNR* set) in which a monomer structure is available that mimics the interaction at atomic level (released prior to 2018). However, when the monomer template does not reproduce the atomic details of the interaction, performance is similar to overall performance in the benchmark (**Figure 5C** and **Supplementary Table 3**). Due to this small number of potential templates (5%) we believe AF2 impressive performance can not be attributed solely to memorization (when available).

2. Additionally, we test the impact of provided templates, as suggested by the reviewer. For our calibration set (*motif* and *non motif* sets) we repeated predictions using the native structures of the complexes as templates. Theoretically, this should have provided the best possible option for prediction. However, we were surprised to find that even if the structure is provided as input, generating alignment separately for the 2 chains, it is not necessarily helpful and does not always improve modeling (**Supplementary Table 4**). In any event, a real world scenario does not supply templates thus we do not report the overall performance using templates. Indeed, during CASP14, the authors of AF2 also discovered that forcing the use of certain templates is not straightforward, and requires additional manipulation of input data, such as down-sampling the MSA. We did not investigate these possibilities further.

3. Examine what happens when you have peptides that should not bind (or try to do an Alanine scanning of the peptide). Does that matter etc? What happens with a poly-A ?

A: We thank the reviewer for this interesting suggestion. Alanine scanning and poly-A docking are now included and discussed in **Figure 3**. AF2 predictions using poly-A peptides fail in all but two cases, emphasizing the importance of the peptide sequence. We also investigated the utility of models for alanine scanning, using Rosetta alanine scanning. We compared the detection of interface hotspots using the native crystal structure to those identified using AF2 models. Our analysis shows that for accurate models (within 2.5Å RMSD) hotspot prediction is highly correlated, with few false positive predictions, but more false negatives, i.e., missed interface hotspots. Moreover, we show that it is also possible to identify interface hotspots by less accurate models, as long as the interface is detected (as we have already reported in a previous study; Marcu et al., Proteins 2017).

4. Run this on a much larger dataset of peptide-proteins with unknown structures (the SLIM database). This might provide some really interesting biological findings.

A: We fully agree with the reviewer that this approach can be used to improve our understanding of regulation via SLIMs. We have here focused on a proof of concept on a large, non-redundant set of peptide-protein complexes (covering a total of over 120 different ECOD families), to show consistent successful modeling of interactions using AF2. Investigation of SLIM-mediated interactions is our current next step to go, and we think that it is out of scope for the present study.

REVIEWERS' COMMENTS

Reviewer #1 (Remarks to the Author):

The authors have significantly improved their manuscript, both in content and english.

I really only have one minor comment: I think that the other related work in BioRxiv should be already mentioned in the introduction and not has a hidden reference later on in the text. Just to put things in proper context as that work actually appeared first in BioRxiv.

Reviewer #2 (Remarks to the Author):

Thanks for the greatly improved manuscript.

The only thing I still miss is a study of if it is possible to separate correct from incorrect cases (i.e. predicting dockQ). I would assume that would be quite easy looking at the surface area (and possible the pLDDT scores).

Reviewer #1 (Remarks to the Author):

The authors have significantly improved their manuscript, both in content and english.

We thank the reviewer for the comments regarding the overall state of the manuscript.

I really only have one minor comment: I think that the other related work in BioRxiv should be already mentioned in the introduction and not as a hidden reference later on in the text. Just to put things in proper context as that work actually appeared first in BioRxiv.

We now mention other work regarding peptide docking with AF2 in the Introduction (now reference #35): In the paragraph preceding the last paragraph therein, we have added the following (highlighted in red):

“We show that by connecting the peptide to the receptor (e.g. by a poly-glycine linker), monomer folding NNs generate accurate peptide-protein complex structures (a similar idea was proposed in parallel by others³⁵).”

Reviewer #2 (Remarks to the Author):

Thanks for the greatly improved manuscript.

The only thing I still miss is a study of if it is possible to separate correct from incorrect cases (i.e. predicting dockQ). I would assume that would be quite easy looking at the surface area (and possibly the pLDDT scores).

We thank the reviewer for the suggestion.

We ran DockQ on the motif and non-motif datasets to evaluate the performance of buried surface area and pLDDT for differentiating between correct and incorrect predictions. This evaluation is now included in Supplementary Figure 7, and referred to in the main text.

We found that when average pLDDT over the peptide residues exceeds 0.7, DockQ values are usually larger than 0.6 as well (representing therefore medium-to high-quality structures). In turn, for buried surface area (calculated with Rosetta Interface analyzer), the results were not as well correlated.

In the Results section, we have added the following text:

“Average pLDDT>0.7 (calculated over peptide residues) is also predominantly associated with high DockQ⁴⁰ values (>0.6) representing medium-to high quality models (This association is stronger than that of normalized Buried Surface Area of models; Supplementary Figure 7).”